# Synergistic Halide- and Ligand-Exchanges of All-Inorganic Perovskite Nanocrystals for Near-Unity and Spectrally Stable Red Emission

**DOI:** 10.3390/nano13162337

**Published:** 2023-08-14

**Authors:** Kaiwang Chen, Dengliang Zhang, Qing Du, Wei Hong, Yue Liang, Xingxing Duan, Shangwei Feng, Linfeng Lan, Lei Wang, Jiangshan Chen, Dongge Ma

**Affiliations:** 1Institute of Polymer Optoelectronic Materials and Devices, State Key Laboratory of Luminescent Materials and Devices, Guangdong Provincial Key Laboratory of Luminescence from Molecular Aggregates, South China University of Technology, Guangzhou 510640, China; chenkw1009@163.com (K.C.); msdlzhang@foxmail.com (D.Z.); duqing326@126.com (Q.D.); yc37821@umac.mo (W.H.); 18335743033@163.com (Y.L.); 202110183327@scut.edu.cn (X.D.); 13060657551@163.com (S.F.); lanlinfeng@scut.edu.cn (L.L.); 2Wuhan National Laboratory for Optoelectronics, Huazhong University of Science and Technology, Wuhan 430074, China; wanglei@mail.hust.edu.cn

**Keywords:** perovskite nanocrystals, ligand assisted reprecipitation, halide-exchange, ligand-exchange, pure red emission

## Abstract

All-inorganic perovskite nanocrystals (NCs) of CsPbX_3_ (X = Cl, Br, I) are promising for displays due to wide color gamut, narrow emission bandwidth, and high photoluminescence quantum yield (PLQY). However, pure red perovskite NCs prepared by mixing halide ions often result in defects and spectral instabilities. We demonstrate a method to prepare stable pure red emission and high-PLQY-mixed-halide perovskite NCs through simultaneous halide-exchange and ligand-exchange. CsPbBr_3_ NCs with surface organic ligands are first synthesized using the ligand-assisted reprecipitation (LARP) method, and then ZnI_2_ is introduced for anion exchange to transform CsPbBr_3_ to CsPbBr_x_I_3−x_ NCs. ZnI_2_ not only provides iodine ions but also acts as an inorganic ligand to passivate surface defects and prevent ion migration, suppressing non-radiative losses and halide segregation. The luminescence properties of CsPbBr_x_I_3−x_ NCs depend on the ZnI_2_ content. By regulating the ZnI_2_ exchange process, red CsPbBr_x_I_3−x_ NCs with organic/inorganic hybrid ligands achieve near-unity PLQY with a stable emission peak at 640 nm. The CsPbBr_x_I_3−x_ NCs can be combined with green CsPbBr_3_ NCs to construct white light-emitting diodes with high-color gamut. Our work presents a facile ion exchange strategy for preparing spectrally stable mixed-halide perovskite NCs with high PLQY, approaching the efficiency limit for display or lighting applications.

## 1. Introduction

With the growing development of high-definition displays, higher requirements were put forward for the luminescent materials in recent years. Semiconductor nanocrystals (NCs) show great application potential in next-generation display technologies due to their good optoelectronic properties, such as narrow full widths at half maximum (FWHM), high-photoluminescence quantum yield (PLQY) and tunable forbidden bandwidth [1,2]. Currently, cadmium chalcogenides are widely used compounds for the preparation of high performance NCs. However, in order to achieve good stability and high PLQY in cadmium chalcogenide NCs, a shell layer needs to be grown on top of the core to provide surface passivation, and the synthesis of core/shell structured NCs usually requires high-reaction temperature [3,4]. In this case, these stable and efficient core/shell NCs show high cost, which is not conducive to large-scale commercial application. Recently, the halide perovskite NCs, particularly the all-inorganic CsPbX_3_ (X = Cl, Br, I), are favored by researchers for their cost-effective synthetic process and outstanding optoelectronic performance [5,6,7]. Similar to the cadmium chalcogenide based NCs, the perovskite NCs can tune their PL peaks by controlling their compositions and crystal sizes to cover the entire visible spectral region. More importantly, the synthesis of perovskite NCs can be accomplished by simpler procedures under mild reaction conditions with low temperatures, and high PLQYs can be obtained without additional shell layers, which enable the perovskite NCs to be more economical for practical applications [8,9,10].

For all-inorganic CsPbX_3_ NCs, X-site engineering is the most facile method to regulate the forbidden bandwidth, because their energy levels of conduction and valence bands are strongly dependent on the properties of Pb-X bonds in the octahedral [PbX_6_]^4−^ [11,12]. And the iodine-dominant CsPbX_3_ NCs can realize emission in the red region. Ordinarily, the PL peak of CsPbI_3_ NCs is located at approximately 700 nm, and can achieve blue-shift by reducing the sizes of NCs [13,14]. But it is still a challenge to synthetize the CsPbI_3_ NCs with pure-red emission matching the requirement of the Rec. 2020 standard, especially at low temperatures. Moreover, the desirable black CsPbI_3_ is usually unstable and easily transforms from the metastable phases (α (cubic), β (tetragonal) or γ (orthorhombic)) to the yellow δ phase at ambient temperature, resulting in greatly reduced luminescence performance [15,16,17,18]. In comparison, the mixed-halide red perovskite NCs of CsPbBr_x_I_3−x_ show better stability because of their more proper tolerance factors, and their emission peaks can be tuned in a larger range by the ratio of iodine and bromine ions [17,19]. At present, the CsPbBr_x_I_3−x_ NCs can be directly synthesized by the conventional methods of hot injection and ligand-assisted reprecipitation (LARP) [20,21]. However, high-performance-pure-red CsPbBr_x_I_3−x_ NCs are rarely achieved, especially by LARP. On the other hand, ion-exchange was proved to be an effective strategy for the preparation of mixed-halide perovskite NCs, which can be carried out at room temperature. Ion-exchange is a post-synthesis treatment that allows the exchange of anions or cations different from the pristine perovskites while essentially inheriting the feature of original crystal structure. By using the ion-exchange, red CsPbBr_x_I_3−x_ NCs can be achieved from high-performance green CsPbBr3 NCs which are more easily synthesized by either hot injection or LARP [12,22,23].

Although the mixture of iodine and bromine ions can approach the desired red emission in perovskites, the mixed halide anions are highly susceptible to segregate into iodine- and bromine-rich domains caused by ion migration [24,25]. In order to inhibit ion migration in the mixed-halide perovskite NCs, the commonly used strategy is to eliminate the ionic defects, which are mainly distributed on the surface [26,27,28]. Generally, Lewis-base ligands, especially the organic amines, have high-binding energy to complex with halides, so they are often used to immobilize the halide ions by forming benign compounds on the surface, and thus reduce the X-site vacancy defects and suppress the halide segregation in the perovskite NCs [29,30]. Furthermore, composition engineering of cations in mixed-halide perovskite NCs is another widely adopted approach to prevent ion migration. For instance, cation doping at the A-site can alter the level of octahedral deformation to effectively reduce ion migration [31,32]. Additionally, the inclusion of Sn at the B-site can improve the photostability of the Pb-based perovskites [33]. Nonetheless, it is still a challenge to achieve B-site doping in perovskite NCs at room temperature.

Hence, we demonstrated the preparation of spectrally stable pure-red CsPbBr_x_I_3−x_ NCs with high PLQY approaching unity by an ion-exchange method. ZnI_2_ was employed as the iodine source to perform the halide-exchange with the green CsPbBr_3_ NCs, which was synthesized by LARP, and the mixed-halide red perovskite NCs were successfully obtained by tuning the ZnI_2_ content. In addition to the halide-exchange, the ligand-exchange was also present during the exchange procedure. The organic ligands were greatly reduced on the surface of the resultant red perovskite NCs, and ZnX_2_ (X = Br and/or I) can be anchored on the surface as inorganic ligands to reduce the vacancy defects of halide anions. Thanks to the synergistic halide- and ligand-exchanges, the mixed-halide red perovskite NCs with organic/inorganic hybrid-ligands realized the stable emission peak at 640 nm with a narrow FWHM of 33 nm, which can match the requirement of the Rec. 2020 standard. The excellent spectral stability should be attribute to the inhibition of halide segregation in the CsPbBr_x_I_3−x_ NCs. More importantly, the PLQY of the pure red perovskite NCs was achieved to be close to 100%, suggesting the significant suppression of non-radiative recombination. This work provides a facile and effective method to prepare efficient and stable red perovskite NCs and paves the way for wide color gamut vivid display applications.

## 2. Experimental Section

### 2.1. Materials

Cs_2_CO_3_ (99.9%) and Octanoic acid (OTAC, 98%) were purchased from Sigma Aldrich (St. Louis, MO, USA). PbBr_2_ (99.99%) was purchased from Xi’an Polymer Light Technology Corporation (Xi’an, China). ZnI_2_ (99.99%) and ZnBr_2_ (99.99%) were purchased from Aladdin (Shanghai, China). Tetraoctylammonium bromide (TOAB, 98%), didodecyldimethylammonium bromide (DDAB, 98%), ethanol amine (99%), Poly(methyl methacrylate) (PMMA), ethyl acetate (99.95%), n-Octylamine (OAm, 99%) and tributylphosphine oxide (TBPO, 98%) were purchased from Macklin Inc. (Shanghai, China). All chemicals were used as received.

### 2.2. Synthesis of CsPbBr_3_ NCs

The green CsPbBr_3_ NCs were synthesized by a LARP method at room temperature [34]. Firstly, the solutions of Cs^+^ (0.05 M Cs_2_CO_3_ dissolved in OTAC) and Pb^2+^ (0.05 M PbBr_2_ and 0.1 M TOAB dissolved together in toluene) were prepared. Then, 280 μL Cs^+^ solution was swiftly added into 2.5 mL Pb^2+^ solution with vigorous stirring for 30 s. Subsequently, 830 μL DDAB solution (12 mg/mL in toluene) was added and stirred for another 5 min to obtain CsPbBr_3_ crude solution. Afterwards, 7 mL ethyl acetate was poured into the CsPbBr_3_ crude solution, followed by centrifuging at 12,000 rpm for 5 min. The supernatant was discarded, and the precipitate was collected and dispersed in 2 mL toluene. After centrifuging at 4000 rpm for 5 min, the solution of CsPbBr_3_ NCs was filtered by the 0.22 μm filter for next experiments and characterization.

### 2.3. Preparation of ZnI_2_ and ZnBr_2_ Precursor Solutions

The precursor solutions of ZnI_2_ were prepared by dissolving ZnI_2_, TBPO and OAm in toluene. The concentration of TBPO was kept at 0.5 M, and the volume ratio of OAm to toluene was maintained at 3%. For preparing the ZnBr_2_ precursors, the process is the same, only replaced ZnI_2_ with ZnBr_2_.

### 2.4. Anion Exchange Processes

A 500 μL ZnI_2_ precursor solution with various ZnI_2_ content was swiftly added into 2 mL solution of CsPbBr_3_ NCs with vigorous stirring for a desired time. Immediately, 7 mL ethyl acetate was poured into the solution following by centrifuging at 12,000 rpm for 8 min. The supernatant was discarded, and the precipitate was dispersed in 2 mL toluene and centrifuged at 5000 rpm for 5 min. Eventually, the resultant solution was filtered by 0.22 μm filter. Notably, the whole process was carried out in a N_2_-filled glove box.

### 2.5. Fabrication of WLED Device

The as-prepared green and red perovskite NCs were mixed with PMMA, respectively. The mixture of the pristine CsPbBr_3_ NCs and PMMA was first coated onto the 460 nm blue LED chip and allowed to dry. Subsequently, the mixture of the red CsPbBr_x_I_3−x_ NCs (0.1 M ZnI_2_ NCs) and PMMA was coated on top of the green layer and then dried again.

## 3. Results and Discussions

The green colloidal CsPbBr_3_ NCs were firstly synthesized by the method of LARP under room-temperature with octanoic acid (OTAC) and didodecyldimethylammonium bromide (DDAB) as the ligands [34], and were dispersed in toluene for the subsequent ion exchange (see the experimental section for details). Figure 1a shows the schematic diagram of the exchange process. Here, n-Octylamine (OAm) from the ZnI_2_ precursor solution plays the role of carriers for halide ions [35,36]. After mixing the colloidal CsPbBr_3_ dispersion and ZnI_2_ precursor, the OAm carried I^−^ can dynamically approach to the surface of the perovskite NCs, and the exchange between Br^−^ and I^−^ will occur spontaneously to form the mixed-halide CsPbBr_x_I_3−x_ NCs (Figure 1a). The PL peaks of the CsPbBr_x_I_3−x_ NCs were precisely tunable between 518 nm and 640 nm by varying the concentration of ZnI_2_ in the precursor solutions (Figure 1b and Appendix A). After the exchange, the spectra of the resultant perovskite NCs remain in a single PL peak with narrow FWHM, indicating the formation of CsPbBr_x_I_3−x_ NCs with the homogeneous phase structure. When the ZnI_2_ concentration is up to 0.08 M and 0.1 M, the CsPbBr_x_I_3−x_ NCs reach the red emission region with PL peaks of 619 nm and 640 nm, respectively. The transmission electron microscope (TEM) measurements were performed to reveal the effect of ZnI_2_ on the sizes and shapes of the CsPbBr_x_I_3−x_ NCs. From the TEM images, it is found that the average size of the red perovskite NCs exhibits an expansion from 12.9 ± 1.9 nm (pristine CsPbBr_3_) to 13.2 ± 1.8 nm (0.08 M ZnI_2_) and 13.7 ± 2.0 nm (0.1 M ZnI_2_), as shown in Figure 1c–e, and their morphologies show ignorable change. Furthermore, the crystallographic characteristics of those perovskite NCs (pristine CsPbBr_3_, 0.08 M ZnI_2_ and 0.1 M ZnI_2_) were studied by the X-ray diffraction (XRD) measurements, and the corresponding XRD patterns are displayed in Figure 1f. The conventional cubic crystal structure is present in all the perovskite NCs, suggesting that the exchange process did not change the crystallographic feature which is consistent with the previous reports [23]. The diffraction peaks of the red CsPbBr_x_I_3−x_ perovskite NCs are consistently shifted to the lower 2θ angles, indicating an increase in cell volume as the radius of I anion is greater than that of the Br anion. These results show that ZnI_2_-dominated anion exchange can achieve the gradual red shift of the PL spectra without destructing the perovskite structure.

The PLQYs of the perovskite NCs were measured for comparison, and the results are shown in Figure 2a and Appendix A. After the exchange of green CsPbBr_3_ NCs with 0.04 M ZnI_2_, the PLQY shows significant reduction from 72.3% to 21.2% and the PL peak changes from 518 nm to 556 nm. Intriguingly, when further increasing the ZnI_2_ concentration, the PLQY of the CsPbBr_x_I_3−x_ NCs is improved gradually with the red shift of PL spectra and approaches unity at 640 nm. The change of PLQY suggests that the suppression of non-radiative loss in the CsPbBr_x_I_3−x_ NCs is strongly dependent on the added ZnI_2_ during the exchange process. To obtain insight into the recombination mechanism in the red perovskite NCs of 0.08 M and 0.1 M ZnI_2_, the time-resolved PLs were measured. In Figure 2b, it is observed that both 0.08 M and 0.1 M ZnI_2_ NCs exhibit bi-exponential decay curves with one fast component and one slow component. The fitted parameters for these decay curves are listed in Appendix A. It is shown that the average lifetime (τ_avg_) of the 0.1 M ZnI_2_ NCs (68.73 ns) is longer than that of the 0.08 M ZnI_2_ NCs (56.62 ns). This suggests that there are fewer nonradiative channels in the 0.1 M ZnI_2_ NCs [37,38], indicating a reduction in defect states of CsPbBr_x_I_3−x_ NCs as the ZnI_2_ concentration increases. Furthermore, the carrier dynamics and the excitons recombination process were investigated by the femtosecond transient absorption (fs-TA) measurements. As shown in Appendix A, the photo-induced bleaching (PB) signals of 0.08 M ZnI_2_ and 0.1 M ZnI_2_ NCs were located at 611 nm and 630 nm, respectively, which are in agreement with the absorption results as shown in Figure 1b. The characteristic decay-associated spectra were then extracted through a global fitting procedure (Figure 2c,d), with τ_1_, τ_2_, and τ_3_ components identified as being responsible for intraband hot-exciton relaxation, exciton trapping to the band gap trap states, and exciton recombination, respectively, as schematically illustrated in Figure 2e [38,39,40]. Results from the global fitting demonstrate that the exciton trapping process of 0.1 M ZnI_2_ NCs was slower compared to that of 0.08 M ZnI_2_ NCs. Additionally, a significant delay in kinetic recombination was observed for 0.1 M ZnI_2_ NCs, which suggests the suppression of non-radiative recombination with an increase in ZnI_2_ precursor solution concentration (Figure 2f).

As described above, the ZnI_2_-based exchange can be used to prepare highly efficient red perovskite NCs. To further study the effect of exchange process on the formation of red perovskite NCs, the in situ PL spectroscopy was employed to monitor the spectral evolution after dropping the ZnI_2_ precursor into the pristine CsPbBr_3_ NCs. Figure 3a,b shows the change of PL spectra during the exchange with 0.1 M ZnI_2_. The PL peak of the perovskite NCs presents negligible redshift during the first second (Figure 3a), suggesting that the halide-exchange is very slight in this period. After then, the PL peak sharply changes to around 600 nm and then gradually redshifts to 640 nm (Figure 3a,b), which clearly indicates that the halide-exchange is very fast at the beginning and then slows. After around 50 s, no change of the PL peak is found in the CsPbBr_x_I_3−x_ NCs, meaning that the equilibrium of halide-exchange is reached. Notably, the PL intensity decreases dramatically during the first second and then gradually increases, and ultimately remains unchanged after around 90 s. The evolution of the PL intensity suggests that the abundant non-radiative channels are quickly formed and then gradually eliminated. When dropping ZnI_2_ precursor, the non-radiative channels could be mainly originated from the rapidly emerged defects located at the surface of the perovskite NCs, leading to the great decrease of PL intensity. We speculate that the surface defects might be caused by the detachment of the original ligands (OTAC and DDAB) and the formation of halogen vacancies. And the abundant halogen vacancies on the surface can trigger the halide-exchange when OAm carried I^−^ approached. Actually, the desorption and absorption of ligands can remain a dynamic balance in the solution of the pristine CsPbBr_3_ NCs. However, this balance is broken by adding the ZnI_2_ precursor, and the ligand-exchange immediately occurs at the surface of NCs. It’s worth noting that the PL intensity continues growing in tens of seconds after the halide-exchange equilibrium. During this period, the reduction of surface defects should be mainly attributed to the effect of ligand-exchange. Finally, the equilibrium of ligand-exchange is achieved and the suppression of surface defects stops, resulting in no change of the PL intensity. More importantly, all the PL spectra in Figure 3b show the single-peak characteristics, implying that the perovskite NCs keeps the homogeneous phase structure during the whole exchange process. And the similar evolution of PL spectra was also observed in the exchange process for the 0.08 M ZnI_2_ NCs, but the establish times for the equilibriums of halide-exchange and ligand-exchange became shorter than those of the 0.1 M ZnI_2_ NCs, as shown in Figure 3c,d.

In the ZnI_2_ precursor solutions, OAm and tributylphosphine oxide (TBPO) were used to improve the solubility of ZnI_2_ in the solvent of toluene. Actually, OAm and TBPO can act as the ligands and would contribute to the exchange process. To study the effect of OAm and TBPO, we performed the in situ PL measurement to probe the emission change of the pristine CsPbBr_3_ NCs after dropping the precursor without ZnI_2_. As shown in Appendix A, the PL intensity of the CsPbBr_3_ NCs also experienced a decrease followed by an increase, which is consistent with the ligand-exchange process as discussed above. But the decrease of the PL intensity is not significant, suggesting that the effect of ligand-exchange on the surface structure of CsPbBr_3_ NCs is limited when adding OAm and TBPO without ZnI_2_. And the absorption, PL, TEM, PLQY, XRD and X-ray photoelectron spectroscopy (XPS) results demonstrate that the exchanged perovskite NCs show the similar characteristics as the pristine CsPbBr_3_ NCs (Appendix A). At the same time, we also studied the effect of Zn^2+^ without I^−^ on the ligand-exchange by adding OAm and TBPO with ZnBr_2_. Appendix A shows the change of the PL spectra after dropping the precursor solution containing 0.1 M ZnBr_2_ into the pristine CsPbBr_3_ dispersion. It is shown that the decrease of PL intensity becomes more significant and faster (in a few seconds), which is accompanied with the blue-shift of PL peaks. This indicates that the presence of Zn^2+^ can encourage the change of surface structure by facilitating the detachment of the initial ligands from the pristine CsPbBr_3_ NCs. And the spectral blue-shift should be attributed to the reduced size of the perovskite NCs, which was confirmed by the TEM result (Appendix A). The extensive detachment of ligands would cause the disruption of the surface lattice and thus decrease the size of the CsPbBr_3_ NCs, and the Zn^2+^, detected by XPS as shown in Appendix A, can combine with Br^−^ to be anchored on the CsPbBr_3_ NCs as the inorganic ligands to “healing” the defective surface, leading to a small increase in the PLQY (Appendix A). The above results demonstrate that the Zn^2+^ plays a very important role during the ligand-exchange process, and its effect on the surface properties of NCs is more effective than OAm and TBPO.

To obtain more insight into the change of ligands between the pristine CsPbBr_3_ NCs and the obtained red CsPbBr_x_I_3−x_ NCs, Fourier transform infrared (FTIR) and XPS measurements were performed. As depicted in Figure 4a, the FTIR spectrum of the pristine perovskite NCs clearly showed the resonances of organic groups (ν_s_ (C-H_x_) = 2700–3000 cm^−1^, ν_s_ (-COOH) = 1680–1700 cm^−1^). The intensities of these peaks greatly dropped for the 0.08 M ZnI_2_ NCs and 0.1 M ZnI_2_ NCs, especially the C=O stretch peaks (ν_s_ (-COOH)), suggesting that the OTAC ligands were mostly removed from the surface of perovskite NCs. From the XPS results (Appendix A), the lowered oxygen contents further verified the reduction of OTAC in the exchanged perovskite NCs. In addition, nitrogen is only present in the ligand of DDAB for the pristine perovskite NCs, and would originate from both of DDAB and OAm for the red NCs. The decrease of N content confirmed the significant reduction of DDAB after exchange (Figure 4b). The pristine CsPbBr_3_ NCs are covered by the ligands of OTAC and DDAB, wherein OTAC is located on the NCs surface through the binding of Pb^2+^ with COO^−^, and DDAB exists on the NCs surface through the interaction of halogen anion and tetra-ammonium cation [34]. One of the main functions of these organic ligands is to passivate the surface defects and thus suppress non-radiative recombination in the perovskite NCs. The reduction of these organic ligands should result in more defects, which act as the non-radiative recombination centers, on the surface of the perovskite NCs. However, the high PLQYs clearly indicate that the non-radiative recombination of the red perovskite NCs is not enhanced with significantly decreasing the amount of organic ligands. It is believed that the Zn^2+^ should make up the loss of organic ligands and thus greatly suppress the non-radiative recombination.

To investigate the role of Zn^2+^, density functional theory (DFT,) simulations were employed to evaluate the impact of Zn^2+^ doping into the lattices of the CsPbBr_3_ and CsPbI_3_ NCs. Our calculations show that the formation energies of doping Zn^2+^ at the B-site of cubic CsPbBr_3_ and CsPbI_3_ are around 0.8 eV (Appendix A). This finding suggests that the doping of Zn^2+^ into the perovskite NCs is an energy absorbing process. However, our exchange reaction was carried out at room temperature and the reaction time was short, which did not allow for the doping of Zn^2+^ into the lattice. Therefore, Zn^2+^ detected by XPS after the anion exchange should be present on the surface of the perovskite NCs as an inorganic ligand, which is consistent with the previous report [41]. The entire exchange processes from the pristine CsPbBr_3_ NCs to the CsPbBr_x_I_3−x_ NCs are illustrated in Figure 4c. After the treatment of CsPbBr_3_ NCs with ZnI_2_ precursors, the exchange reaction can be divided into three steps. Firstly, the addition of ZnI_2_ precursor disrupts the balance of ligands on the pristine green CsPbBr_3_ NCs, and massive desorption of the original ligands results in the destruction of lattice structure at the NCs surface, leading to abundant surface defects, including halogen vacancies. Secondly, the formed defective surface triggers the rapid halide-exchange to produce CsPbBr_x_I_3−x_ NCs; simultaneously, the ligand-exchange is taking place, and the Zn^2+^ combined with halogen ions can be anchored as the inorganic ligands to passivate the surface defects. Finally, the halide-exchange slows and then reaches an equilibrium followed by the ligand-exchange equilibrium in the CsPbBr_x_I_3−x_ NCs.

Mixed halide perovskites are known to undergo photo-induced halide segregation, which will result in the spectral instability [42,43,44]. In order to study the photostability of the red CsPbBr_x_I_3−x_ NCs, we measured the PL spectra of the perovskite films under intense illumination of ultraviolet light. A simplified schematic of the measurement equipment is shown in Figure 5a. The PL spectra and their peak intensities of the 0.1 M ZnI_2_ NCs did not exhibit any change after 5 h of irradiation, indicating the absence of phase separation caused by photo-induced halide segregation, as depicted in Figure 5b. Encouraged by the exceptional photostability and high PLQY of the obtained red perovskite NCs, we prepared the white light-emitting diode (LED) by combining the 0.1 M ZnI_2_ NCs and pristine green CsPbBr_3_ NCs with a commercial blue LED chip. In Figure 5c, it is evident that there are three distinct emission peaks which are assigned to the corresponding blue chip, green CsPbBr_3_ NCs and red 0.1 M ZnI_2_ NCs. Notably, there is no intermediate emission peak detected, indicating that the perovskite NCs were effectively separated in the PMMA matrix to block the anion exchange between the green and red NCs. Figure 5d displays the CIE chromaticity coordinates of the blue chip, pristine green CsPbBr_3_ NCs and 0.1 M ZnI_2_ NCs. The selected CIE color coordinate triangle, including the corresponding blue, green and red monochromatic emission, encompasses 120% of the National Television System Committee’s (NTSC) latest standard (Rec. 2020). This demonstrates that these green and red perovskite NCs, which are easily obtained by LARP and ion-exchange, respectively, can be used as the promising low-cost emitters in wide color gamut display devices.

## 4. Conclusions

In this study, we synthesized efficient and spectrally stable red CsPbBr_x_I_3−x_ NCs by a facile ion-exchange technique of post-processing CsPbBr_3_ NCs with ZnI_2_ in solution. The in situ PL measurements were conducted to probe the exchange procedures. It was observed that the halide-exchange will rapidly arise and then gradually decelerate to achieve an equilibrium. And the original organic ligands were found to partially desorb from the surface of the perovskite NCs to trigger the halide-exchange in a very short time, and simultaneously facilitate the ligand-exchange. More importantly, zinc halides can be anchored on the surface of perovskite NCs as the inorganic ligands to reduce the vacancy defects of halide anions formed during the ion-exchange process, which is beneficial to stabling the emission spectra. The synergistic effect of the halide-exchange and ligand-exchange enabled the mixed-halide red perovskite NCs with the hybrid organic/inorganic ligands to achieve a stable emission peak at 640 nm with a narrow FWHM of 33 nm and a PLQY close to unity. Ultimately, we combined the exchanged red perovskite NCs and the pristine green perovskite NCs with a commercial blue LED chip to fabricate the white device with wide color gamut and high-color saturation. Our study presents a cost-effective strategy to develop efficient and stable perovskite NCs for applications in full-color displays.

## Figures and Tables

**Figure 1 nanomaterials-13-02337-f001:**
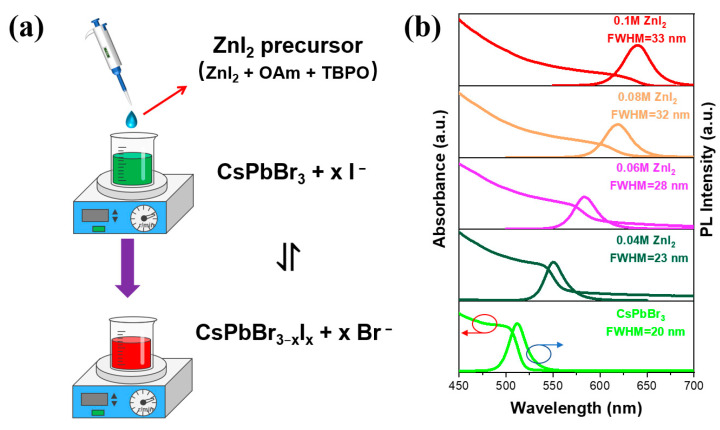
(**a**) Schematic illustration of the anion exchange in toluene. (**b**) UV-vis absorption (red arrow) and PL spectra (excitation at 365 nm, blue arrow) of the pristine CsPbBr_3_ NCs and the CsPbBr_x_I_3−x_ NCs after anion-exchange with different ZnI_2_ concentration in precursor solution. Size histogram of the perovskite NCs: (**c**) pristine CsPbBr_3_, (**d**) 0.08 M ZnI_2_ and (**e**) 0.1 M ZnI_2_ (inset: TEM images of the perovskite NCs). (**f**) XRD patterns of the perovskite films prepared from the pristine CsPbBr_3_, 0.08 M ZnI_2_ and 0.1 M ZnI_2_ NCs.

**Figure 2 nanomaterials-13-02337-f002:**
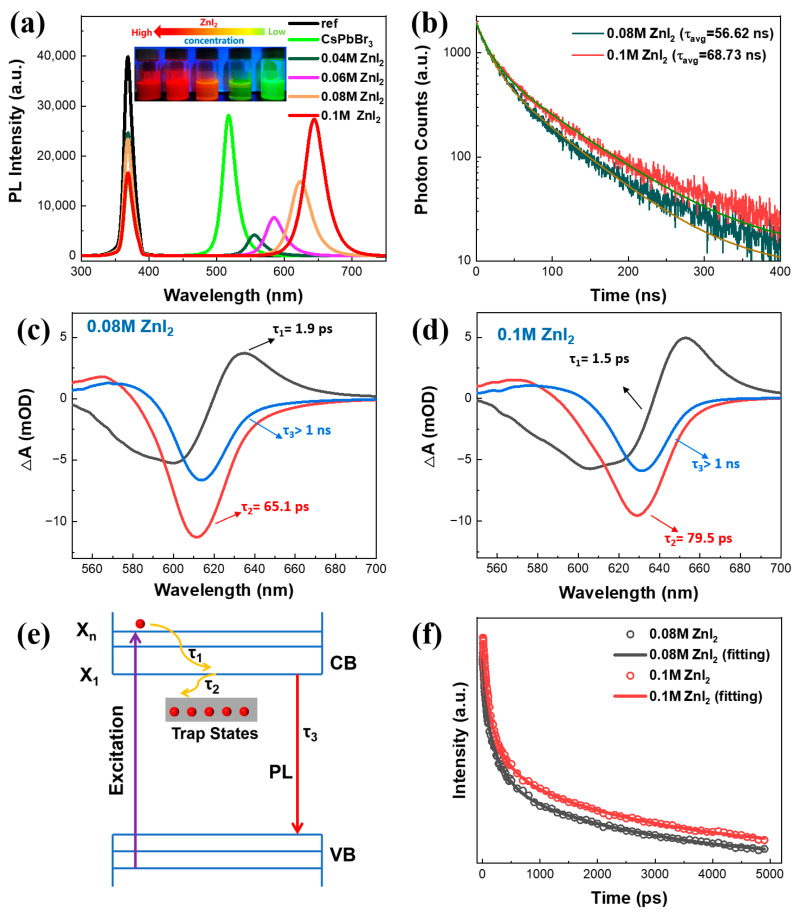
(**a**) PLQY spectra of the pristine CsPbBr_3_ NCs and the CsPbBr_x_I_3−x_ NCs treated with different ZnI_2_ concentrations. Insets are the photographs of the corresponding perovskite NCs under the irradiation of UV lamp. (**b**) PL decay curves for the samples of 0.08 M ZnI_2_ NCs and 0.1 M ZnI_2_ NCs. Characteristic decay-associated spectra for (**c**) the 0.08 M ZnI_2_ NCs and (**d**) the 0.1 M ZnI_2_ NCs. (**e**) Schematic illustration of the photoinduced relaxation processes involved in the perovskite NCs. (**f**) Bleaching recovery kinetics for 0.08 M ZnI_2_ NCs and 0.1 M ZnI_2_ NCs.

**Figure 3 nanomaterials-13-02337-f003:**
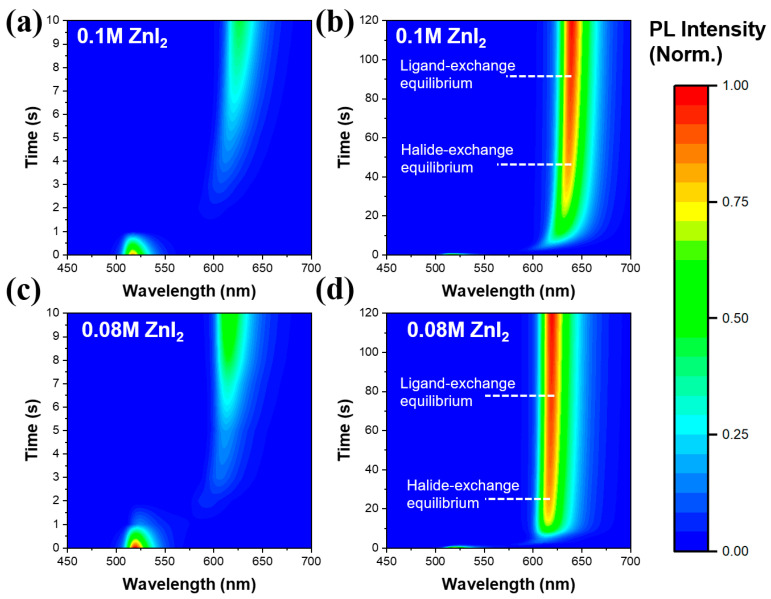
PL evolution during the exchange process by adding ZnI_2_ precursor into the as-prepared CsPbBr_3_ NCs. The fitted 2D PL spectra for the 0.1 M ZnI_2_ NCs (**a**,**b**) and the 0.08 M ZnI_2_ NCs (**c**,**d**). Color scale bar is provided on the right side for the intensity of the PL.

**Figure 4 nanomaterials-13-02337-f004:**
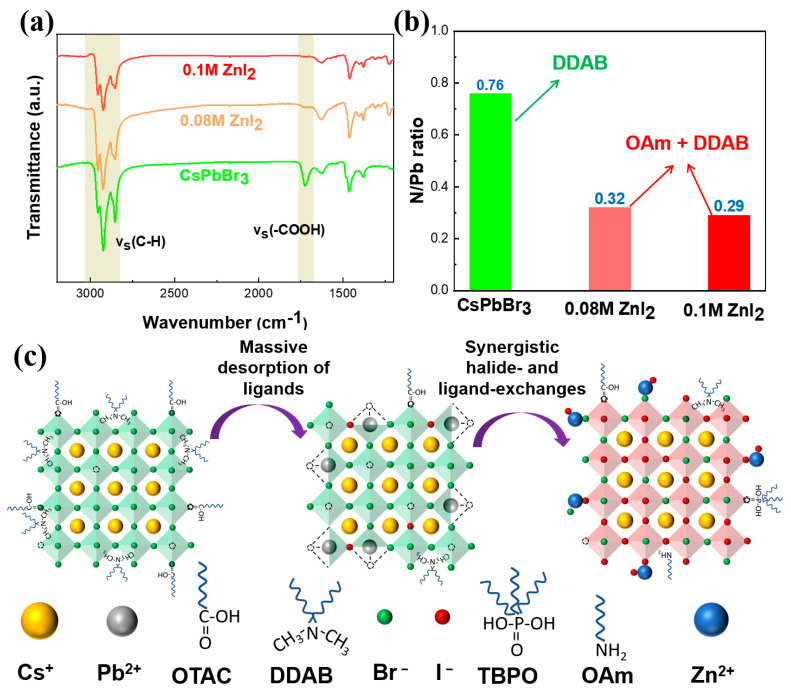
(**a**) FTIR spectra of the pristine CsPbBr_3_ NCs and the 0.08 M and 0.1 M NCs. (**b**) N/Pb atomic ratios in the perovskite NCs calculated from the XPS data. (**c**) Schematic diagram of the entire exchange process from the green CsPbBr_3_ NCs to the red CsPbBr_x_I_3−x_ NCs.

**Figure 5 nanomaterials-13-02337-f005:**
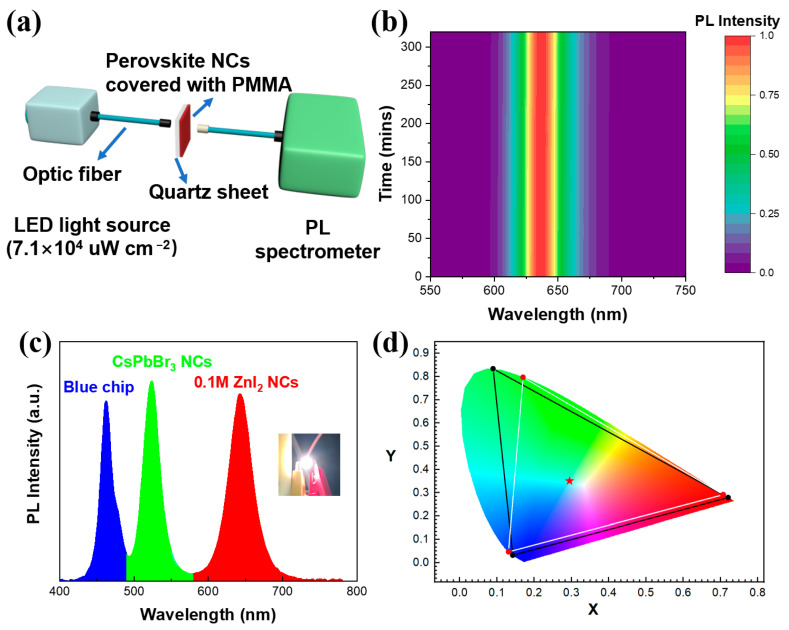
(**a**) Schematic of the spectrometer equipment used to characterize the photostability of the perovskite NCs. (**b**) The fitted 2D PL spectra of the 0.1 M ZnI_2_ NCs over time (under the illumination of 365 nm LED with a power of 7.1 × 104 μW cm^−2^). (**c**) PL spectra of the white LED (inset: the photograph of the device operated at 2.8 V). (**d**) Color triangle of the blue LED, pristine green CsPbBr_3_ NCs and 0.1 M ZnI_2_ NCs (black line) compared to the NTSC TV standard (white line) and Color coordinate of the white LED (red star).

## Data Availability

The data that support the findings of this study are available from the corresponding author upon reasonable request.

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
