# Peer review of "Synergistic Halide- and Ligand-Exchanges of All-Inorganic Perovskite Nanocrystals for Near-Unity and Spectrally Stable Red Emission"

_nanomaterials, 2023, doi:10.3390/nano13162337_

Round 1

Reviewer 1 Report

The manuscript is aimed to  synthesis and spectroscopy of efficient and spectrally stable red CsPbBrxI3-x NCs by a 361 facile ion-exchange technique of post-processing CsPbBr3 NCs with ZnI2 in solution.

There are several lacks in interpretation occur:

1) What is nature of the trap states in studied materials? Are these traps formed during synthesis or due to exciton decay?

2) In the figure 2b the experimental points are not well resolved multiexponential curve. Probably, the other longer components exist. Authors should measure the decay curve in wider time range. From the other hand, this curve could be well-fitted using hyperbola equation.

3) The picosecond luminescence could be attributed to the entire exchange process. More probably the excitation quenching on the shallow traps

Reviewer 2 Report

All-inorganic lead halide perovskites are attractive materials for displays. The authors report synthesis and characterization of CsPbBr3-xIx nanocrystals with PLQY near unity. The results are very interesting and the discussion sound. I strongly recommend acceptance of this paper for publications but I also have a few minor remarks. Firstly, halide exchange occurs within the nanocrystals but ligand-exchange is on the surface. I suggest to changes slightly the abstract, for instance changing “CsPbBr3 NCs are first synthesized….” by “CsPbBr3 NCs with surface organic ligands are first synthesized….”. Secondly, for me as a chemist, Zn2+ (lines 311-313) or ZnI2 are not ligands but halide anions are. I recommend to rewrite some sentences to make this clear. Thirdly, there are many sentences starting with “and” or “but”. They should be corrected to avoid “and” and “but “ as the first words. Fourthly, in lines 151 and 152 “-“ at Br and I should be as superscripts. 

Reviewer 3 Report

The manuscript entitled “Synergistic halide- and ligand-exchanges of all-inorganic perovskite nanocrystals for near-unity and spectrally stable red emission” by Chen et al. presents a comprehensive research work carried out on the perovskite nanocrystals of CsPbBrxI3-x by ion-exchange. A systematic analysis of structure and PL properties of nanocrystals is presented. The WLED device presented in this study is interesting for the researchers working in the field of organic LEDs especially with perovskite devices.

The article is very well written and the experimental section is presented with enough details.

 I have no hesitation to accept this manuscript. This research work will attract the attention of the research community of Nanomaterials.

Author Response

Dear reviewers:

  Thank you for thoroughly reviewing our article and providing such a positive feedback. We greatly appreciate the time and effort you put into the review.

                                                                                            Sincerely.